

# 1 Global Gridded Crop Model evaluation:
# 2 benchmarking, skills, deficiencies and
# 3 implications

*Christoph Müller* [1], *Joshua Elliott* [2,3], *James Chryssanthacopoulos* [3], *Almut Arneth* [4], *Juraj*
*Balkovic* [5,6], *Philippe Ciais* [7], *Delphine Deryng* [2,3], *Christian Folberth* [5,8], *Michael Glotter* [9], *Steven*
*Hoek* [10], *Toshichika Iizumi* [11], *Roberto C. Izaurralde* [12,13], *Curtis Jones* [12], *Nikolay Khabarov* [5],
*Peter Lawrence* [14], *Wenfeng Liu* [15], *Stefan Olin* [16], *Thomas A. M. Pugh* [4,17], *Deepak Ray* [18],
*Ashwan Reddy* [12], *Cynthia Rosenzweig* [3,19], *Alexander C. Ruane* [3,19], *Gen Sakurai* [11], *Erwin*
*Schmid* [20], *Rastislav Skalsky* [5], *Carol X. Song* [21], *Xuhui Wang* [7,22], *Allard de Wit* [10], *Hong*
*Yang* [15,23]
[1] *Potsdam Institute for Climate Impact Research, 14473 Potsdam, Germany*
[2] *University of Chicago and ANL Computation Institute, Chicago, IL 60637, USA*
[3] *Columbia University Center for Climate Systems Research and NASA Goddard Institute for Space Studies, New York,*
*NY 10025, USA*
[4] *Karlsruhe Institute of Technology, IMK-IFU, 82467 Garmisch-Partenkirchen, Germany*
[5] *International Institute for Applied Systems Analysis, Ecosystem Services and Management Program, 2361*
*Laxenburg, Austria*
[6] *Comenius University in Bratislava, Department of Soil Science, 842 15 Bratislava, Slovak Republic*
[7] *Laboratoire des Sciences du Climat et de l'Environnement. CEA CNRS UVSQ Orme des Merisiers, F-91191 Gif-sur-*
*Yvette, France*
[8] *Department of Geography, Ludwig Maximilian University, 80333 Munich, Germany*
[9] *University of Chicago, Department of the Geophysical Sciences, Chicago, IL 60637, USA*
[10] *Alterra Wageningen University and Research Centre, Earth Observation and Environmental Informatics, 6708PB*
*Wageningen, Netherlands*
[11] *National Agriculture and Research Organization, National Institute for Agro-Environmental Sciences, Agro-*
*Meteorology Division, Tsukuba, 305-8604, Japan*
[12] *University of Maryland, Department of Geographical Sciences, College Park, MD 20742, USA*
[13] *Texas A&M University, Texas AgriLife Research and Extension, Temple, TX 76502, USA*
[14] *National Center for Atmospheric Research, Earth System Laboratory, Boulder, CO 80307, USA*
[15] *Eawag, Swiss Federal Institute of Aquatic Science and Technology, CH-8600 Duebendorf, Switzerland*
[16] *Department of Physical Geography and Ecosystem Science, Lund University, 223 62 Lund, Sweden*
[17] *School of Geography, Earth & Environmental Science and Birmingham Institute of Forest Research, University of*
*Birmingham, Edgbaston, Birmingham, B15 2TT, United Kingdom*
[18] *Institute on the Environment, University of Minnesota, Saint Paul, USA*
[19] *National Aeronautics and Space Administration Goddard Institute for Space Studies, New York, NY 10025, USA*
[20] *University of Natural Resources and Life Sciences, Institute for Sustainable Economic Development, 1180 Vienna,*
*Austria*
[21] *Rosen Center for Advanced Computing, Purdue University, West Lafayette, Indiana, USA*
[22] *Peking University, Sino-French Institute of Earth System Sciences, 100871 Beijing, China*
[23] *Department of Environmental Sciences, University of Basel, Petersplatz 1, CH-4003 Basel, Switzerland*



*Correspondence to: Christoph Müller (Christoph.Mueller@pik-potsdam.de)*





## Abstract

Crop models are increasingly used to simulate crop yields at the global scale, but there so far is no general framework on how to assess model performance. We here evaluate the simulation results of 14 global gridded crop modeling groups that have contributed historic crop yield simulations for maize, wheat, rice and soybean to the Global Gridded Crop Model Intercomparison (GGCMI) of the Agricultural Model Intercomparison and Improvement Project (AgMIP). Simulation results are compared to reference data at global, national and grid cell scales and we evaluate model performance with respect to time series correlation, spatial correlation and mean bias. We find that GGCMs show mixed skill in reproducing time-series correlations or spatial patterns at the different spatial scales. Generally, maize, wheat and soybean simulations of many GGCMs are capable of reproducing larger parts of observed temporal variability (time series correlation coefficients (r) of up to 0.888 for maize, 0.673 for wheat and 0.643 for soybean at the global scale) but rice yield variability cannot be well reproduced by most models. Yield variability can be well reproduced for most major producer countries by many GGCMS and for all countries by at least some. A comparison with gridded yield data and a statistical analysis of the effects of weather variability on yield variability shows that the ensemble of GGCMs can explain more of the yield variability than an ensemble of regression models for maize and soybean, but not for wheat and rice. We identify future research needs in global gridded crop modeling and for all individual crop modeling groups. In the absence of a purely observation-based benchmark for model evaluation, we propose that the best performing crop model per crop and region establishes the benchmark for all others, and modelers are encouraged to investigate how crop model performance can be increased. We make our evaluation system accessible to all crop modelers so that also other modeling groups can test their model performance against the reference data and the GGCMI benchmark.



## 1. Introduction

Agriculture is fundamental to human life and our ability to understand how agricultural production responds to changes in environmental conditions and land management has for long been a central question in science (Russell, 1966; Spiertz, 2014). Numerical crop models have been developed over the last half-century to understand agricultural production systems and to predict effects of changes in management (e.g. irrigation, fertilizer) (El-Sharkawy, 2011). In the face of continued population growth, economic development, and the emergence of global-scale phenomena that affect agricultural productivity (most prominently climate change) crop models are also applied at the global scale (Rosenzweig and Parry, 1994). Given the importance of climate change and the central interest in agriculture, global-scale crop model applications have been increasingly used to address a wide range of questions, also beyond pure crop yield simulations (e.g., Bondeau et al., 2007; Del Grosso et al., 2009; Deryng et al., 2014; Osborne et al., 2013; Pongratz et al., 2012; Rosenzweig et al., 2014; Stehfest et al., 2007; Wheeler and von Braun, 2013).

With very few exceptions, crop models applied at the global scale have been developed for field-scale applications (e.g. EPIC-based models, pDSSAT, pAPSIM) or have been derived from global ecosystem models by incorporating field-scale crop model mechanisms and parameters (e.g. LPJ-GUESS, LPJmL, ORCHIDEE-crop, PEGASUS) and several of these have been systematically intercompared with a large number of other field-scale models (Asseng et al., 2013; Bassu et al., 2014). Still, differences between global gridded crop models (GGCM) (Rosenzweig et al., 2014) and also between field scale models (Asseng et al., 2013; Bassu et al., 2014; Li et al., 2015) have been recently identified, following a general call to revisit modeling skills and approaches (Rötter et al., 2011), which is also a central objective of the Agricultural Model Intercomparison and Improvement Project (AgMIP) (Rosenzweig et al., 2013) and the Inter-Sectoral Impact Model Intercomparison Project (ISI-MIP) (Warszawski et al., 2014). Site-specific applications and model evaluation can demonstrate the general suitability of the mechanisms implemented in the models and the corresponding parameters (Boote et al., 2013), but the extrapolation and upscaling of parameters and model assumptions remains challenging (Ewert et al., 2011; Hansen and Jones, 2000). If models are applied at the global scale, they also need to be assessed at the scale of interpretation, which ranges from gridded to national or regional aggregates (Elliott et al., 2014a; Fader et al., 2010; Müller and Robertson, 2014; Nelson et al., 2014a; Nelson et al., 2014b; Osborne et al., 2013).

Global-scale applications of crop models face a number of challenges. A major difference to field-scale model applications is that at large regional to global scale detailed model calibration to field observations is not possible. Specification and initialization as typically conducted in field-scale applications simply lack data of suitable spatial coverage and simulation units (e.g. 0.5° grid cells) represent an aggregate of many smaller, potentially heterogeneous fields. Initialization of soil properties (Basso et al., 2011) is especially important in dry and nutrient-depleted production systems (Folberth et al., 2012) and the specification of soil properties can greatly affect crop model simulations (Folberth et al., 2016). Similarly, production systems typically cannot be specified in great detail. There is limited information on growing seasons (Portmann et al., 2010; Sacks et al., 2010) and irrigation area, amount and timing (Siebert et al., 2015; Thenkabail et al., 2009) that can be used to model crop-specific irrigation shares (Portmann et al., 2010; You et al., 2010), planting dates and crop parameters for the specification



of varieties grown (van Bussel et al., 2015) and multiple cropping rotation practices. Still, crop varieties
are often assumed to be homogeneous globally or within large regions in global model setups (Folberth
et al. in prep., Müller and Robertson, 2014). Other management aspects are typically assumed to be
static in space and time. There have been some attempts to calibrate crop models in global-scale
applications but these always calibrate to (sub-)national yield statistics (Fader et al., 2010) or to gridded
yield data sets (Deryng et al., 2011; Sakurai et al., 2014) that are based on (sub-)national statistics (Iizumi
et al., 2014b; Mueller et al., 2012).
The evaluation of model performance (skill) faces similar challenges. Data availability has improved
lately, as gridded data sets on yield time series have become available (Iizumi et al., 2014b; Ray et al.,
2012), but generally only yield data is available, while other end-of-season (e.g. biomass) or within-
season (e.g. leaf area index, LAI) information is lacking. The gridded yield data sets are not purely
observational but include some form of model application in the interpolation of unknown accuracy so
that they do not directly qualify as a reference data set. Currently, global gridded crop models lack a
clear benchmark against which they can be evaluated. A benchmark is an a-priori definition of expected
model performance based on a set of performance metrics (Best et al., 2015). Given that the GGCMs are
merely driven by variable information on weather and atmospheric $CO_2$ concentrations whereas
assumptions on soil properties and/or management systems are static, these cannot be expected to
reproduce all temporal dynamics and spatial patterns of observed crop yields. The contribution of
weather variability has been estimated to roughly one third globally of the observed yield variability (Ray
et al., 2015) and moderate-to-marked yield losses can be explained by weather data over 26-33% of the
harvested area (Iizumi et al., 2013), with a clear negative impact of extreme drought and heat events
(Lesk et al., 2016). The explanatory power of weather variability on crop yields varies strongly between
regions, with a tendency to have larger influence on yield variability in high-input systems than in low-
input systems (Ray et al., 2015), where substantial variation may also be introduced by pests and
diseases, socio-economic conditions, and changes in management.
The comparison with gridded data is difficult, because of introduced interpolation errors in the
referenced data. The differences between the two gridded yield reference data sets can be substantial,
indicating that the modeling assumptions made introduce substantial uncertainty and limit their
applicability as a reference data set. Similarly, if simulated gridded yield data are to be compared with
(sub-)national yield statistics, these need to be spatially aggregated. This aggregation requires
information on the spatial and temporal distribution of cropland and irrigation systems, which is
available from different global data sets with differing estimates that can introduce substantial
uncertainty (Porwollik et al., under review).
The objective of this paper is to provide and discuss a broad model evaluation framework to test
performance of GGCMs that participated in the global gridded crop model intercomparison (GGCMI) of
AgMIP's Gridded Crop Model Initiative (Ag-GRID) (Elliott et al., 2015). We aim to assess general and
individual model performance across different crops and regions that can serve as a basis for further
model development and improvement as well as a benchmark for future assessments. Model
performance is evaluated with respect to correct spatial patterns as well as temporal dynamics at the
global scale as well as for individual countries and grid cells. Reference data sets and metrics are



explained in more detail in the methods section. We also propose this evaluation system to become a
standard benchmarking system for all global gridded crop model application and to track model
improvement[1]. As such, we make the data processing and the computation of performance metrics
available online to other modelers so that they can compare their models' results against the GGCMI
ensemble. We argue that under given uncertainties the best performing crop model per region and crop
defines the benchmark for the other models.

---

[1] We are currently setting up an online evaluation system where files can be uploaded and assessed in the same way as the GGCMI simulations in this paper. The tool will become available on the GEOSHARE Portal at https://mygeohub.org/groups/geoshare





## 2. Methods

### 2.1. Models participating and experimental setup

For the GGCMI in AgMIP, 14 model groups have contributed (Table 1), following the protocol for the
GGCMI (Elliott et al., 2015). For this, crop modeling groups were asked to perform global simulations
with their standard assumptions (inputs or internal calculations) on growing seasons and fertilizer inputs
('*default*'), with harmonized growing seasons (i.e with supplied planting and harvest dates (Elliott et al.,
2015)) and fertilizer inputs per crop and pixel ('*fullharm*') as well as a simulation with harmonized
growing seasons but assuming the absence of nutrient limitation ('*harm-suffN*', referred to as 'harmnon'
in Elliott et al. (2015), but changed here to avoid the misinterpretation of "no nitrogen"). We evaluate
model performance for each of these harmonization sets to study the importance of these assumptions
for individual models' as well as for the ensemble's performance. More detail on the processes
implemented in the GGCMs can be found in the supplement, tables S1-S4.
We here use data from simulations by these 14 GGCMs driven by the weather data set AgMERRA (Ruane
et al., 2015), for which all modeling groups have performed simulations and historical atmospheric
carbon dioxide ($CO_2$) concentrations (Thoning et al., 1989). The AgMERRA data set spans the time frame
of 1980-2010 and provides daily data on the most important meteorological driver variables and groups
applied their own interpolation to sub-daily values if needed. If additional weather data were needed by
individual modeling groups (such as long-wave radiation), these were supplemented from the Princeton
Global Forcing data set (PGFv2) (Sheffield et al., 2006). We assume this to have little impact on
simulation results, as all data sets are based on station data and/or reanalysis data and as bias-correction
of re-analysis data is performed for each meteorological variable individually, there is no explicit
dependency between individual variables (e.g. between radiation and temperature). The contribution of
uncertainties in historic weather data sets on crop model skill is evaluated elsewhere (Ruane et al., in
prep.) and is not part of the objectives here.
All input and harmonization targets are supplied at a regular grid with 0.5 degree resolution. Weather
data are supplied at daily resolution. Each modeling group is asked to use their own soil data and
parameterization (Elliott et al., 2015). Yield simulations are conducted for the four major crops wheat,
maize, rice and soy depending on model capacities. Some groups could not supply data for all crops or
harmonization settings (see Table 2). Each modeling group supplied data for each crop for all land grid
cells (up to 62911 grid cells) with separate simulations for purely rain-fed conditions and for conditions
with full irrigation. Full irrigation does not necessarily imply the absence of water stress in all models, if,
e.g. the atmospheric water vapor pressure deficit exceeds the plant's physical capacity to transpire
water. Model irrigation is triggered on demand (supplement Table S2) independent of the availability of
irrigation water (Elliott et al., 2015).
Following FAO reporting standards, we are not reporting simulated yield data as calendar aggregates but
as a time series of annual growing seasons. In this way, we avoid that individual calendar years can have
two harvests (one shortly after January $1^{st}$ and one shortly before December $31^{st}$) and others with zero
harvest, which would greatly increase the variability in the reported simulated crop yields and would be
inconsistent with FAO data. Instead, each harvest season is assigned a calendar year, starting with the



first harvest of the growing season that started in 1980 (beginning of the AgMERRA forcing data), leaving
a residual uncertainty how the time series need to be matched (see below).

## 2.2.  Reference data

We use two different data sets for the evaluation of the GGCMs. The FAO data (FAOstat data, 2014) is
used for national and global-scale model evaluation and is available at these scales from 1961-2013. For
some countries, production data and/or harvested areas have been estimated by the FAO rather than
reported (FAOstat data, 2014). For spatially resolved detail we use the data published by Ray et al. (2012,
henceforth "Ray2012"), as that allows for direct comparison with the regression model analysis of Ray et
al. (2015, henceforth "Ray2015"). The Ray2012 data spans 1961-2008 and was aggregated from its
original resolution of 5 arc minutes to the 0.5° GGCMI standard resolution, weighted by production. Both
production and harvested area data are collected at sub-national level for 51 countries in the Ray data
and changes in productivity thus reflect both dynamics in area and production. National totals are forced
to match FAO statistics, if there were differences (Ray et al., 2012). The assignment of yield statistics to
the grid raster as conducted by Ray et al. (2012) requires making assumptions that introduce
uncertainty. To illustrate the uncertainty in the gridded reference data, we compare the Ray2012 data
with the Iizumi data set (Iizumi et al., 2014b). The Iizumi data set is available in gridded form from 1982-
2006, which we here re-gridded from its original resolution of 1.125°x1.125° to the standard GGCMI
resolution of 0.5°x0.5° resolution, using the remapcon function (CDO, 2015). As much of the southern
hemisphere has no data for 2006 due to its ending in the middle of Southern summer, we only consider
the period 1982-2005 here. The Iizumi data are based on national FAO data and the spatial variability
within countries is introduced based on satellite data. Given the different approaches, there are
substantial differences in spatial patterns between the Ray and Iizumi data, but temporal dynamics at
the national level reflect the FAO data.

## 2.3.  Metrics used:

In the analysis we largely focus on time series correlation of simulated and reference crop yields, given
that the main application of gridded crop models at the global scale is related to studies on climate
change impacts, where we expect models to respond reasonably to changes in atmospheric conditions
(weather, climate). The main metric used is therefore the time series correlation analysis, employing the
Pearson's product moment correlation coefficient (henceforth "correlation coefficient"). Significance
levels (p-values) are reported based on a t-distribution with length(x)-2 degrees of freedom. Given
difficulties in attributing sequences of growing periods to the calendar year in both FAO statistics[2] and in
simulated data where groups also interpreted the reported standards differently, we test if the time
series correlation can be substantially improved by shifting the times series by one year. We apply such
shifts only if the correlation coefficient improves by at least 0.3 and report un-shifted time series
analyses in the supplement. Time series correlation is used at the global aggregation level, the national
aggregation and the pixel level. In some cases, the correlation analysis is weighted by production to put
higher emphasis on larger production units, assuming that data quality is often better than for smaller

---

[2] FAO glossary on crop production: „… When the production data available refers to a production period falling into two successive calendar years and it is not possible to allocate the relative production to each of them, it is usual to refer production data to that year into which the bulk of the production falls." Available at http://faostat3.fao.org/mes/glossary/E





producer units (e.g. less developed countries) and because these are more important to correctly
simulate for global assessments. At the global scale, correlation coefficients are simply reported in the
figures but we employ heatmaps to display correlation coefficients at the national scale, making use of a
version of the heatmap.2 function of the gplot package (Warnes et al., 2016), which has been modified
to allow for extra labeling.
We acknowledge that the models are only driven by fields of weather data, soil data and nitrogen
fertilizer inputs, ignoring the heterogeneity in patterns of other fertilizers (e.g. P, K), pest control and
other managerial aspects (e.g. varieties, planting densities). Therefore, we only test model performance
in reproducing spatial patterns of productivity at national aggregations and not within individual
countries, as the quality of gridded reference data Ray2012 (interpolated (sub-)national statistics) as well
as fertilizer inputs (Elliott et al., 2015; Mueller et al., 2012) and growing seasons (Elliott et al., 2015;
Portmann et al., 2010; Sacks et al., 2010) is limited with respect to the spatial heterogeneity. Deviations
from national or global yield levels are computed as the mean bias, as in eq. 1, where *i* is any element in
*n*. At the global scale and for individual countries, *n* is the number of growing seasons in the sample.
$bias = \frac{1}{n}\sum_{i=1}^{n}(yield_{sim,i} - yield_{obs,i})$                                      eq. 1
For a more comprehensive testing of the simulated yield dynamics, we employ Taylor diagrams that
allow for displaying the correlation in spatio-temporal patterns between observations and simulated
data in a single diagram (Taylor, 2001). The Taylor diagram depicts the correlation coefficient across
spatial units and time, the centered RMSD, and the variance relative to that of the observational data
set. Acknowledging the difficulties with respect to the spatial heterogeneity in reference and simulated
data, we employ the Taylor diagrams only for nationally aggregated data, meaning that spatial patterns
only refer to national aggregations here. In the Taylor diagram analysis, countries are weighted by their
crop-specific production (FAOstat data, 2014). To disentangle the contribution of the spatial vs. the
temporal variability to the Taylor diagram, we also compute two variants of these diagrams which focus
on temporal or spatial variability only. For the temporal-dynamics-only variant, we remove the national
means from all de-trended time series so that all national time series have a mean of zero and thus
display no differences in this respect. For the space-dynamics-only variant, we average time series so
that we compute the metrics with one national mean value per country only, ignoring possible changes
in data quality over the time series. For plotting Taylor diagrams, we use the taylor.diagram function of
the R package plotrix (Lemon, 2006) that we have modified to allow for weighted correlation and for
testing of significance levels.
Instead of numerous maps on pixel-specific performance metrics, we also present these in form of
boxplots. To allow for weighting the distribution of pixel-specific metrics such as the correlation
coefficients, we employ weighted quantiles of the function quantileWt of the R package simPopulation
(Alfons and Kraft, 2013).

### 2.4. Data processing

Gridded crop model simulations are driven by time series of weather data and of atmospheric $CO_2$
concentrations, and static management assumptions. A comparison to observation-based reference data



thus requires processing of raw simulation GGCM outputs and the reference data to make these
different data sources comparable. As much of the trends in yield are driven by intensification and
altered management (FAO, 2013; Ray et al., 2012), we are removing trends from simulation and
reference data. As reference data are available at grid-cell, national and global levels, we aggregated
simulated yield data to grid-cell, national, and global levels, using an area-weighted average as described
in eq. 2. Aggregation to the grid-cell level only describes the combination of irrigated and rain-fed
simulation time series, but follows the same principle.
$$yield_{aggregated,t} = \frac{\sum_{i=1}^{n} yield_{i,ir,t} * area\_irrigated_{i,t} + \sum_{i=1}^{n} yield_{i,rf,t} * area\_rainfed_{i,t}}{\sum_{i=1}^{n} (area\_irrigated_{i,t} + area\_rainfed_{i,t})}$$ eq. 2
Here, $i$ is the index of any grid cell assigned to the spatial unit in question for growing season $t$, $n$ is the
number of grid cells in that spatial unit, $yield_{i,ir,t}$ is the simulated yield (t/ha) under fully irrigated
conditions in grid cell $i$, and $yield_{i,rf,t}$ is the simulated yield (t/ha) under rain-fed conditions in grid cell $i$,
$area\_irrigated_i$ is the irrigated harvested area (ha) in grid cell $i$ and $area\_rainfed_i$ is the rain-fed harvested
area (ha) in grid cell $i$.
Following Porwollik et al. (under review), we use four different masks for the aggregation to national
data: MIRCA2000 (Portmann et al., 2010), SPAM (You et al., 2014a; You et al., 2014b), Iizumi (Iizumi et
al., 2014b), and Ray (Ray et al., 2012). As we cannot assess which of these aggregation masks is superior
to the others, we always select the aggregation mask that gives the best agreement between simulated
and reference time series. MIRCA2000 and SPAM provide separate data on irrigated and rain-fed crop-
specific harvested areas per grid-cell, while Ray and Iizumi do not distinguish irrigated from rain-fed
areas. For aggregation purposes, we thus separate total harvested area per grid cell and crop from Ray
and Iizumi into irrigated and rain-fed areas, using the relative shares per grid cell and crop from
MIRCA2000 (see Porwollik et al., under review).
After aggregation to national time series or to grid-cell specific area-weighted combinations of irrigated
and rain-fed yield simulations, we remove trends from simulated and reference data. For this, we are
computing the anomalies by subtracting a moving mean average of a 5-year window (t-2 to t+2), with 3-
year windows at both ends (t1- to t+1) of the time series in order to not lose too many years from the
time series. Similar de-trending methods have been applied by other studies (Iizumi et al., 2014a; Iizumi
et al., 2013; Kucharik and Ramankutty, 2005). We also tested other de-trending methods (e.g. linear or
quadratic trend removal) and find that this may also results in better agreement between simulated and
reference data sets. However, for simplicity we focus on one de-trending method only in this analysis.
For evaluation across different countries, de-trended time series can be compared as pure anomalies,
which vary around zero, or with preserved national mean yields allowing also for assignment of
differences in yield levels between different countries.
For a comparison of simulated yields that are reported in t/ha dry matter with FAOstat yields (FAOstat
data, 2014), which are reported in t/ha "as purchased", we assume a net water content of 12% for maize
and wheat, 13% for rice and 9% for soybean, following Wirsenius (2000). This assumption does not affect
any metrics other than the mean bias.



## 2.5.  Benchmarks for evaluating model performance

GGCM simulations are typically used to study effects of changing environmental conditions, such as climate change impact assessments. We therefore put much emphasis on the models' ability to reproduce temporal variability. Also the spatial variability of crop yields, e.g. along environmental gradients within countries or in response to different fertilizer input within and between countries should be reproduced by the models.

We apply weights when assessing model performance. For analyses of aggregated yield data, it is important to get large areas and highly productive areas right in the simulations. Also, reference data is often of limited quality for marginal and/or small areas. We therefore typically weight results by production (harvested area multiplied with productivity).

At pixel scale, we are presenting skill-based model ensemble estimates by selecting the single best GGCM per pixel that demonstrate the joint ensemble skill rather than an average (e.g. median) across all models. This skill-based approach demonstrates to what extent crop models can actually reproduce observed patterns and variability and differences between individual models and the skill-based model ensemble quantify the learning potential within the ensemble. Principally, in the absence of other benchmark measures, the best performing model should be the benchmark for the others. For the definition of the benchmark here, we do not only consider the GGCMI ensemble but also the 27 regression models as used by Ray et al. (2015). A model-based benchmark as postulated here can establish a very low target, e.g., if all models perform poorly. As such, the benchmark will have to be continuously re-assessed and model intercomparison studies as the GGCMI can help to further develop this benchmark.



## 3. Results

We present results from the evaluation for three different aggregation levels: global, national and grid-
cell level. The global level is the most aggregate where underlying reasons for observed patterns are
hard to identify. National-level data provides more insights on underlying patterns but requires data
reduction for presentation. Pixel-level results can only be assessed by statistical means and results are
thus presented in aggregated form again. We typically display results for the *default* setting in the main
text but supply results for all other settings in the supplement. For the main text figures, we use *fullharm*
simulations for all those model/crop combinations that did not supply a *default* setting simulation (i.e.
those that did not have a default setting before participating in GGCMI). These are clearly indicated in
figures and captions. Also, to reduce the amount of data displayed here, we typically show results for
maize in the main text and display figures for all other crops in the supplement, while still describing and
discussing these here.

### 3.1. Global scale model performance

Aggregated to global time series of crop yields, the different GGCMs display mixed skill in comparison to
the FAOstat time series when both are de-trended. Of the four major crops, global yield variability can be
best reproduced for maize with correlation coefficients (r) between 0.89 and 0.42 and one non-
significant correlation (PRYSBI2, Figure 1). PRYSBI2 is actually parametrized to reproduce the historic
trend in crop yields and if trends are not removed prior to the time series correlation analysis, its
correlation becomes highly significant with a correlation coefficient of 0.56. Note that a correlation
analysis that includes a trend to which the model has been calibrated may be strongly dominated by this
trend. Changes in the harmonization setting (*fullharm*, *harm-suffN*, see Figures S1 and S2 in the
supplement) often have little effect on simulations except for a few models, where harmonization can
significantly improve (e.g. EPIC-BOKU) or weaken (e.g. PEGASUS) the correlation.
For wheat, 10 of the 14 models produce a time series that is significantly correlated to FAO statistics
(Figure 2) with correlation coefficients between 0.67 and 0.37. Harmonization does not greatly change
correlation coefficients but 2 models achieve significant correlation under harmonization that they did
not achieve in the *default* setting (GEPIC, ORCHIDEE-crop) whereas one loses the significant correlation
under harmonization (PEGASUS, see Figures S3-S4). PRYSBI2 again only achieves significant correlation if
trends are not removed prior to the correlation analysis.
Only 3 of the 11 GGCMs that submitted data for rice (Table 2) achieve significant correlation to FAO
statistics of variations in global rice productivity (EPIC-IIASA, LPJ-GUESS and PRYSBI2, Figure 3) and two
other achieve significant correlations under *fullharm* (EPIC-BOKU, PEPIC, Figure S5), but none of the
models reaches statistical significance under the *harm-suffN* setting (Figure S6). PRYSBI2's correlation
improves substantially (from 0.53* to 0.83***) if trends are maintained.
Of the 13 GGCMs that submitted data for soybean (Table 2), 7 achieve significant correlation to FAO
statistics of variations in global soybean productivity (correlation coefficients between 0.64 and 0.41).
Under harmonization, two more models reach statistical significance levels (LPJ-GUESS, PEPIC, figures S7-
S8) and PRYSBI2 reaches significant correlations (0.57**) if trends are not removed.



There are also great differences between GGCMs concerning their absolute deviation from observed
yield levels, reflecting their different setups, process representation and calibration (Table S2-S4 in the
supplement). We find no relationship between mean bias and the ability to reproduce variability over
time (time series correlation) for maize (Figure 5), wheat (Figure S9) and rice (Figure S10) but a positive
relation (that is, correlation coefficients tend to be higher for larger mean bias) was found for soybean
(Figure S11).
## 3.2. National scale
National aggregated yield data is presented as time-series correlation coefficients (color-coded in
heatmaps) as well as the mean bias. We here only show the top-ten producer countries for maize and
display data for the other crops and for all producer countries in the supplement.
Inter-annual variability of most top ten maize producer countries can be reproduced to large extent by
various GGCMs. The inter-annual variability of Indonesia cannot be reproduced well by any of the
models (max r is 0.5 and correlation is not statistically significant in most cases), whereas the inter-
annual variability of Argentina, France, India, South Africa and the United States can be largely
reproduced by almost any GGCM-harmonization combination. To achieve good statistical correlations,
some time series had to be shifted by a year, especially for Argentina, Mexico and South Africa (Figure
S12). Also for the other maize producer countries, the yield variability can be well reproduced by most
GGCM-harmonization settings, and there is always at least one GGCM that can reproduce a statistically
significant share of the variability (Figure S13).
For wheat (Figures S14-S16), rice (Figures S17-S19) and soybean (Figures S20-S22) a similar picture
emerges. The yield variability of the top 10 producer countries can be reproduced by a large number of
GGCMs, with a few exceptions (France and China for wheat; Bangladesh and Myanmar for rice; China for
soybean) where only a few GGCMs are able to reproduce statistically significant shares of the yield
variability in the FAO yield statistics. Likewise for wheat, rice and soybean, a statistically significant share
of the yield variability can be reproduced for all producer countries covered here (best column in Figures
S16, S19, S22) and allowing for shifts in the time series can greatly improve the correlation, especially in
tropical countries (e.g. Pakistan for wheat, Indonesia and Thailand for rice, soybean in India).
Other than deviating in temporal dynamics, which is tested with time-series correlation analyses, GGCM
simulations can also be biased compared to FAO yield statistics, typically underestimating yields in high-
yielding countries and overestimating yields in low-yielding countries (Figure 7). Some GGCMs (e.g.
pDSSAT) and the *harm-suffN* generally tend to overestimate yields, but not in all cases (Figures 7, S23-
S26).
Aggregation to national scale does not only allow for looking into temporal dynamics of each individual
country, it also allows for assessing spatial patterns in combination with temporal dynamics. By
assembling national yield data series to a 2-dimensional field (countries x time), we can assess the
spatio-temporal correlation between simulated and FAO data as well as the variance and centered RMSD
using Taylor diagrams (Taylor, 2001). Here, countries are weighted by production (FAOstat data, 2014) to
avoid that small countries dominate the overall picture (see Methods). GGCMs show mixed skill when
compared to FAO data, with some models having high correlation coefficients, whereas others have low



or negative correlation coefficients (Figure 8). Here, *harm-suffN* simulations typically show much lower
correlation coefficients than the other harmonization settings. Except for one model under *harm-suffN*
(EPIC-TAMU, Figure 8), harmonization (*fullharm*, *harm-suffN*) eliminates any negative correlation
coefficients. None of the GGCM-harmonization settings leads to negative correlation coefficients if the
national differences in mean yields are ignored (Figure S28). The Taylor diagram with flattened time
dimension (i.e. only using one multi-annual mean per country in the analysis, Figure S27) almost looks
identical to the Taylor diagram with both the time and space dimension (Figure 8). This disentangling of
the contributions of spatial vs. temporal variability shows that the overall skill of models as presented in
the Taylor diagram is dominated by the spatial signal, i.e. the differences between national mean yields
outweigh the year-to-year variability around those means by far. This also explains why GGCMs with
some calibration against yield levels (EPIC-IIASA, LPJmL, PEGASUS, PRYSBI2, see table S4) show relatively
high correlation coefficients, as the differences between national means dominate the overall
correlation. When the spatial differences are ignored by removing the mean yields per country (i.e. each
country has a mean of zero and the correlation thus only considers the year-to-year variability around
these), the GGCMs perform more similar, typically displaying correlation coefficients between 0.4 and
0.6 (Figure S28) and often the variance becomes larger (larger standard deviation) relative to the FAO
reference data set.
A similar pattern can be observed for the other crops as well. The differences in yield levels between
countries dominate the overall performance in the spatio-temporal correlation (Figures S29 vs. S30 for
wheat, S32 vs. S33 for rice, S35 vs. S36 for soybean) and GGCMs perform more similar in the analysis of
time-only variance (Figures S31, S34, S37).

### 3.3.   Pixel scale

At the pixel scale, reference data uncertainty increases substantially, as the two available data sets are
essentially model- and observation-based interpolations of (sub-)national yield statistics, and neither of
the two is independent from FAO national data. Differences between the two gridded yield reference
data sets (Iizumi et al., 2014b; Ray et al., 2012) are expressed via a time series correlation analysis after
removing trends via a moving average (see Methods, Figure 9).
Independent of the harmonization setting, the GGCMI model ensemble (selecting the best correlation
per pixel across the different GGCMs and harmonization settings) finds statistically significant
correlations ($p<0.1$) with Ray2012 in most of the currently cropped areas for all four crops analyzed here
(Figure 10 for maize, Figures S38 – S40 for wheat, rice and soybean). The spatial patterns with high
correlations are comparable to where Ray2015 could find significant influence of weather on crop yield
variability with an ensemble of 27 regression models, but the GGCMI ensemble finds statistically
significant contributions of weather (the only dynamic driver in the model simulations) over a much
larger area than Ray2015. The original analysis of Ray2015 could find better correlations for large parts
of China, the Corn Belt in the USA and individual countries in Africa, most notably Kenya and Zimbabwe.
Contrary to the GGCMI ensemble (best per pixel), individual GGCMs find statistically significant
correlations in a much smaller area, largely comparable to the 27 regression model ensemble used by
Ray2015, see e.g. pDSSAT simulations for maize in the supplement (Figure S41). There is no eminent
pattern in the performance of individual GGCMs and none of the GGCMs performs in any region



significantly better than all others (see e.g. Figure S42 for best performing GGCM per grid cell for maize
under the default setting).
Some individual GGCMs achieve similar distribution of correlation coefficients with the gridded maize
yield data set of Ray2012 as the ensemble of the 27 regression models as used by Ray2015, but most
perform less well (Figure 11). As at the global-scale and national-scale aggregation level, harmonization
can improve or worsen GGCM performance, depending on the GGCM.
For wheat, the GGCMI ensemble also finds statistically significant correlations for a much larger area
than the regression model ensemble used by Ray2015, but correlation coefficients are often lower (e.g.
in Europe) even though the spatial patterns with relatively high correlations coefficients are similar
between the GGCMI ensemble and those reported by Ray2015 (see Figure S38). As for maize, the
harmonization has little effect on the ensemble skill. Also the distribution of coefficients of
determination values shows that GGCMs can reach higher values for individual pixels but are generally
(individually and as the total ensemble) less well correlated with the gridded Ray data set than the 27
regression models of Ray2015, see Figures S38 and S43.
A similar picture emerges for rice, where also Ray2015 only find low correlation coefficients, whereas the
GGCMI ensemble covers a much broader area and finds moderate correlation coefficients in South
America, India and Australia, but not in China as Ray2015 does. As for wheat, individual GGCMs can
reach higher coefficients of determination values than the regression model ensemble of Ray et al.
(2015) for individual pixels, but generally the correlations found are weaker than for the regression
model ensemble as used by Ray2015, see Figures S39 and S44.
For soybean, the GGCMI ensemble also covers a broader range than the regression model ensemble
used by Ray2015. As for maize, the GGCMI ensemble finds equally high correlation coefficients as the
regression model ensemble, with the notable exception of western Russia (Figure S40). Soybean yield
variability in the USA can be better reproduced by the GGCMI ensemble than by the regression models
employed by Ray2015. Again, some individual GGCMs perform equally well as the regression model
ensemble employed by Ray2015, whereas the GGCMI ensemble achieves better coefficients of
determination than the regression model ensemble used by Ray2015 (Figure S45). Also here, some
GGCMs profit from harmonization, whereas others have better performance under their default setting
or are not sensitive to the harmonization at all.



## 4. Discussion

### 4.1. Benchmark: What to expect from GGCMs

It is implausible to expect crop models to reproduce vast shares of yield variability and spatial patterns of crop yields given their coarse resolution, reliance on static inputs, and reliance on weather data when this is but one driver of true yield variability. This is particularly true for low-input regions where many other elements such as unsuitable management or pest outbreaks may contribute substantially to yield variability. It is questionable if the statistical analysis of Ray2015 should define the expectations for crop model performance as their regression models are driven with rather aggregate weather information (precipitation and temperature of either the growing season or of the 12 month preceding harvest). As GGCMs often find stronger influence of weather variability than Ray2015, especially for maize and soy, it is plausible to assume that weather variability is at least as important as described by Ray2015. On the other hand, regression models can be derived from many time series and as none of the GGCMs can reproduce the strong influence of weather variability on crop yields as e.g. reported for maize in Kenya or soybean in Russia (Ray et al., 2015), these strong relationships may be statistical artifacts or based on other weather-related dynamics that are not captured by the GGCMs, such as weather-related pest outbreaks (e.g., Esbjerg and Sigsgaard, 2014). Similar considerations apply for national and global-scale performance. However, also here it can be generally expected that weather variability is more important for yield variability in countries with high-input agriculture than in low input countries. GGCM simulations should not be expected to reproduce yield variability of countries that do not directly report production and harvested area to the FAO and where data gaps are filled with FAO estimates (Folberth et al., 2012).

Gridded crop models make a number of simplifications, such as homogeneous management across larger areas, including soils, sowing dates and varieties. Within individual farming regions, sowing varies by days to even weeks as sowing dates are subject to a number of weather-induced conditions (e.g. soil wetness, soil temperature) and the timely availability of labor and machinery and farmers may chose different varieties to grow. The mixture of management practices within regions thus buffers observed variability in the region's yield records, as the diversity should cancel out the variability to some extent when aggregated to a region average. GGCMs on the contrary implement highly homogeneous systems that tend to overestimate variability, allowing for no or little variation in sowing dates across the years or within larger regions (Sacks et al., 2010) and assuming no change in crop varieties across the simulation period of 31 years. This variety selection does not only contribute to the technology-driven trend in crop yields, which we have removed here (see Methods), but may also alter the crops' response to adverse environmental conditions. The model simplifications also encompass simplified assumptions on the distribution of fertilizers and varieties, which should not only affect the temporal dynamics simulated but also the spatial patterns of crop yields.

### 4.2. GGCM performance

Maize and soybean are the crops where the GGCMs show the best skill in reproducing reference data variability, followed by wheat and rice. The separation of temporal and spatial variability shows that the spatial variability dominates the overall variability in data simply because the differences between national yields are typically greater than those between individual years within countries. GGCMs that



perform some level of calibration against national data therefore score relatively high in correlation
coefficients (e.g. Figure 8) but not necessarily for greater model skill as the national differences are
imposed in the calibration process. If nutrients are assumed to be non-limiting (*harm-suffN*), the
reproduction of spatial patterns is reduced and these simulations (orange symbols in e.g. Figure S27) are
therefore typically less extreme in comparison to the *default* settings (blue in e.g. Figure S27) and closer
to the analysis of only temporal dynamics (e.g. Figure S28). Harmonization of management assumptions
affects only in some cases the time-series correlation in individual countries (e.g. Figure 6). Simulations
with no nutrient limitation typically lead to a greater mean bias in yield simulations (e.g. Figure 7) but not
necessarily to large changes in time series correlation, suggesting that calibration or mean biases often
do not affect the model's skill to respond to interannual variation in weather conditions. However, it also
often leads to greater variance in the time series (orange symbols move outwards relative to blue
symbols in Figures S28, S31, S34, S37). The effect of harmonization is not only dependent on the
individual GGCM's sensitivity to these assumptions but also to the difference between the *default* and
the harmonized settings with respect to growing season and fertilizer input.
For maize and soy, the GGCMI ensemble outperforms an ensemble of 27 regression models (Ray et al.,
2015) with respect to area with significant correlation and to correlation coefficients (Figures 11 and
S45), indicating that model performance is good. As there are still regions in which GGCMs are
outperformed by the regression models (e.g. Kenya for maize, Russia for soybean), and because the
individual GGCMs show varying skill for different regions, each of the models has sufficient room for
improvement if we consider the best performing model is the benchmark for all others.
For wheat, GGCMs show less influence of weather variability than Ray2015 and should thus strive to
achieve similar performance levels as the regression models used by Ray2015. The simulation of wheat is
complicated by the mixture of spring and winter wheat varieties that are also grown within the same
regions and where the current distinction in the models and the GGCMI growing season data may not be
accurate. For future analyses, we therefore recommend to perform separate simulations for spring and
winter wheat.
Rice is generally not simulated with great skill by any GGCM or the overall ensemble. However, also the
regression model ensemble of Ray2015 does not detect substantial influence of inter-annual weather
variability in much of the rice growing areas, suggesting that rice production systems are currently not
well represented in GGCMs and also cannot be captured well by regression models. Possible causes
could be the complexity of the multiple cropping seasons in rice production (Iizumi and Ramankutty,
2015) and the assumptions on irrigation, which is especially in rice production which is largely irrigated.
There is considerable uncertainty in historic weather patterns, as reflected by the 9 different weather
data products used in GGCMI. We here use only one of these weather data sets for which all GGCMs
submitted data with different management scenarios (*default*, *fullharm* [harmonized growing periods
and nutrient inputs], *harm-suffN* [harmonized growing periods with no nutrient stress]). The differences
between weather products and their effects on GGCMs' skill to reproduce observed time-series
variations are discussed in more detail in Ruane et al. (in prep.).



### 4.3.  Data processing and assumptions

There are a number of caveats with respect to the processing of data. We employ a moving average
approach to remove trends from observation-based and simulated data. There are various other
methods to remove trends from time series (e.g. linear or quadratic trends) which we have tested as
well. No clear picture has emerged to what method is best as this is dependent on the individual time
series. We argue that the most important aspect in this de-trending is that observation-based and
simulated data are treated in the same way. Also, the moving average seems to be least dependent on
assuming an underlying functional form as e.g. linear or quadratic de-trending methods and thus is more
robust across the broad range of yield time series (global, national, grid cells). Data aggregation is based
on global data sets on harvested areas per crop. Porwollik et al. (under review) have demonstrated that
this can greatly affect results for individual crop x GGCM x country combinations. We here chose to use
the best matching aggregation mask in each case, arguing that as long as none of the harvested area
data sets can be excluded for quality concerns all are equally plausible and their disagreement should
not be held against the crop models.
We find that shifting time series by a year can sometimes greatly improve the correlation between
simulated and reference time series, e.g. converting a non-significant correlation into a highly significant
(p<0.01) correlation with high correlation coefficients (r=0.89) for LPJ-GUESS *harm-suffN* maize
simulations for South Africa or converting negative correlation coefficients (r < -0.5) to positive (r > 0.5)
for PEGASUS *fullharm* maize simulations in China (Figures 6 and S12). We acknowledge that some of this
is owing to the relatively vague definition of how FAO yields are attributed to calendar years and how
this matches with assumed growing periods in the GGCM simulations. However, this seems to be an
important improvement to be achieved by future global crop modeling studies. The GGCMI phase I
protocols request that data are reported as a series of growing season harvests (Elliott et al., 2015)
rather than calendar years to avoid complications with harvest year attribution if harvest occurs around
the end of the calendar year. Moreover, years are removed from the record if sowing occurred during
the spinup, i.e. part of the growing season is not within the supplied weather input. Data reporting of
future GGCMI simulations will have to be improved to better enable a direct matching of simulated and
reference time series. If time series correlation at the global scale could be improved by time shifts,
obviously the correlation would be even more improved, if individual country time series would have
been adjusted as needed before aggregation rather than shifting the aggregated time series. However,
this is beyond the scope of the study here.

### 4.4.  Implications for future crop model development and analyses

Further model development and improvement is needed in collaboration with field-scale modeling
approaches (Asseng et al., 2013; Bassu et al., 2014; Li et al., 2015) and experimentalists (Boote et al.,
2013). Improvements are also wanted for the representation and aggregation of soils in GGCM
simulations (Folberth et al., 2016) and management including growing season data and fertilizer types,
amounts and timing (Hutchings et al., 2012). But also information on soil management, crop varieties,
crop rotations, and actual irrigation amounts and schemes is presently not or only incompletely available
and better information could greatly inform global crop modeling. Scrutinizing underlying reasons (e.g.
the detail on management considered in the simulations) for good or poor model performance is,
however, beyond the capabilities of this study and the individual modeling groups are requested to



investigate their model's strengths and weaknesses. The overall model evaluation and the GGCMI phase
I modeling data set (Elliott et al., 2015) enable such analyses but cannot be conducted centrally. The
work by Folberth et al. (in prep.) is a good example of how the underlying reasons for differences in
model performance can be identified for individual crop models.
Also, yield statistics in themselves are not a good reference data set for dissecting model functionality as
errors in various processes such as gross primary production, respiration, allocation of photosynthate,
soil dynamics and crop stress response can compensate each other in the formation of yield. Site data
measurements do not only provide data on targeted experiments (as e.g. the FACE experiments, see e.g.
Leakey et al. (2009)) but also on related water and carbon dynamics, as e.g. eddy flux tower
measurements that can help to get good simulation results for good reasons. As such it remains crucial
to also test global-scale models against detailed data from experiments to build trust in the underlying
mechanisms. This point-scale evaluation of models has been performed for several of the GGCMs
engaged here and is not subject of this study (e.g., Gaiser et al., 2010; Izaurralde et al., 2006; Jones et al.,
605    2003).

We propose that future global or large-scale gridded crop models are tested against the GGCMI model
ensemble and the reference data used here to establish a benchmark for model evaluation and future
model development. This cannot overcome the shortage in suitable reference data, but it provides a first
benchmark against which global gridded crop models can be tested. We are well aware of the
shortcomings to establish a benchmark that largely consists of modeled data (Best et al., 2015; Kelley et
al., 2013), either from other models or from model-assisted interpolation of highly aggregated statistics
but see no other option under current data availability. Also, the benchmark should not be confused
with a validation of models, but establishes a reference point against which model performance can be
evaluated. We here assume that the best performing model currently defines the model performance
that can be expected, but acknowledge that the underlying reasons for good (and poor) model
performance need to be better understood in order to avoid defining statistical artifacts as a benchmark
for models.

## 5. Conclusions

Agricultural productivity is increasingly modeled at the global scale, but model setup and evaluation is
hampered by the lack of high-quality input and reference data. We establish a first global crop modeling
benchmark using a crop model ensemble of 14 crop modeling groups and reference data at grid cell,
national and global scale. Even though crop models often demonstrate good performance in reproducing
temporal and spatial patterns of observed crop yields, there is also the need to improve all models. We
argue that the value of the crop model ensemble in an intercomparison study is the ability to learn from
each other as models often show complimentary skill. We encourage all future crop model development
to be tested against the GGCMI global crop model benchmark and thus make our evaluation framework
publicly accessible at https://mygeohub.org/groups/geoshare. This modeling intercomparison exercise
provides a benchmark for facilitating model improvements by the individual modeling groups. There is
substantial crop modeling skill for the simulation of maize, wheat and soybean yields at the global scale,
but rice simulations are currently not preforming well and will require additional effort to improve these



simulations. Ongoing collaboration with field-scale modelers and experimentalists is needed to improve
model mechanisms and parameters. Finally our results emphasize the need for continuous development
and improvement of detailed agricultural data for model input and model evaluation that cover the
entire global agricultural land.



## Code availability

The code of the processing scripts is available via github at https://github.com/RDCEP/ggcmi

The evaluation pipeline will be made available at https://mygeohub.org/groups/geoshare after publication of the paper.

## Data availability

Model output data will be made available via the GGCMI data archive.

## Author contribution

CM and JE designed the experiment and the evaluation framework in discussion with all co-authors. CM, JE, and JC developed the code for the evaluation and data processing. CM, JE, JB, DD, CF, SH, RCI, CJ, NK, PL, WL, SO, TAMP, ARe, GS, EW, RS, XW and AdW performed model simulations. TI and DR provided reference data. CXS developed the online tool for model evaluation. CM wrote the manuscript with contributions from all co-authors.

## Competing interests

We declare non competing interests.

## Acknowledgements

We acknowledge the support and data provision by the Agricultural Intercomparison and Improvement Project (AgMIP). CM acknowledges financial support from the MACMIT project (01LN1317A) funded through the German Federal Ministry of Education and Research (BMBF). AA and TAMP were supported by the European Commission's 7th Framework Programme under Grant Agreement number 603542 (LUC4C) and by the Helmholtz Association through its research program ATMO.



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



**Table 1: GGCMs participating in the study, model type and key references.**

| Crop model | Model type | Key literature |
|---|---|---|
| CGMS-WOFOST | Site-based process model | de Wit and van Diepen (2008) |
| CLM-Crop | Ecosystem Model | Drewniak et al. (2013) |
| EPIC-BOKU | Site-based process model (based on EPIC) | EPIC v0810 - Izaurralde et al. (2006); Williams (1995) |
| EPIC-IIASA | Site-based process model (based on EPIC) | EPIC v0810 - Izaurralde et al. (2006); Williams (1995) |
| EPIC-TAMU | Site-based process model (based on EPIC) | EPIC v1102 - Izaurralde et al. (2012) |
| GEPIC | Site-based process model (based on EPIC) | EPIC v0810 - Liu et al. (2007); Williams (1995); Folberth et al. (2012) |
| LPJ-GUESS | Ecosystem Model | Lindeskog et al. (2013); Smith et al. (2001) |
| LPJmL | Ecosystem Model | Waha et al. (2012), Bondeau et al. (2007) |
| ORCHIDEE-crop | Ecosystem Model | Wu et al. (2015) |
| pAPSIM | Site-based process model | APSIM v7.5 - Elliott et al. (2014b); Keating et al. (2003) |
| pDSSAT | Site-based process model | pDSSAT v1.0 - Elliott et al. (2014b); DSSAT v4.5 - Jones et al. (2003) |
| PEGASUS | Ecosystem model | v1.1 - Deryng et al. (2014), v1.0 - (Deryng et al., 2011) |
| PEPIC | Site-based process model (based on EPIC) | EPIC v0810 - Liu et al. (2016), Williams (1995) |
| PRYSBI2 | Empirical/process hybrid | Sakurai et al. (2014) |





**Table 2: Data availability by GGCM, crop and harmonization setting. Crosses (X) indicate availability, dashes (-) indicate that**
**data was not supplied. The three columns per crop are the different harmonization settings on management (*default*,**
***fullharm* and *harm-suffN*, see above).**

| GGCM | Maize | | | Wheat | | | Rice | | | Soybean | | |
|---|---|---|---|---|---|---|---|---|---|---|---|---|
| | Default | fullharm | harm-suffN | default | Fullharm | harm-suffN | default | fullharm | harm-suffN | default | fullharm | harm-suffN |
| CGMS-WOFOST | X | - | - | X | - | - | X | - | - | X | - | - |
| CLM-Crop | X | X | X | X | X | X | X | X | X | X | X | X |
| EPIC-BOKU | X | X | X | X | X | X | X | X | X | X | X | X |
| EPIC-IIASA | X | X | X | X | X | X | X | X | X | X | X | X |
| EPIC-TAMU | - | X | X | - | X | X | - | - | - | - | - | - |
| GEPIC | X | X | X | X | X | X | X | X | X | X | X | X |
| LPJ-GUESS | X | - | X | X | - | X | X | - | X | X | - | X |
| LPJmL | X | - | X | X | - | X | X | - | X | X | - | X |
| ORCHIDEE-crop | X | X | X | X | X | X | X | X | X | X | X | - |
| pAPSIM | X | X | X | X | X | X | - | - | - | X | X | X |
| pDSSAT | X | X | X | X | X | X | X | X | X | X | X | X |
| PEGASUS | X | X | X | X | X | X | - | - | - | X | X | X |
| PEPIC | X | X | X | X | X | X | X | X | X | X | X | X |
| PRYSBI2 | X | - | - | X | - | - | X | - | - | X | - | - |





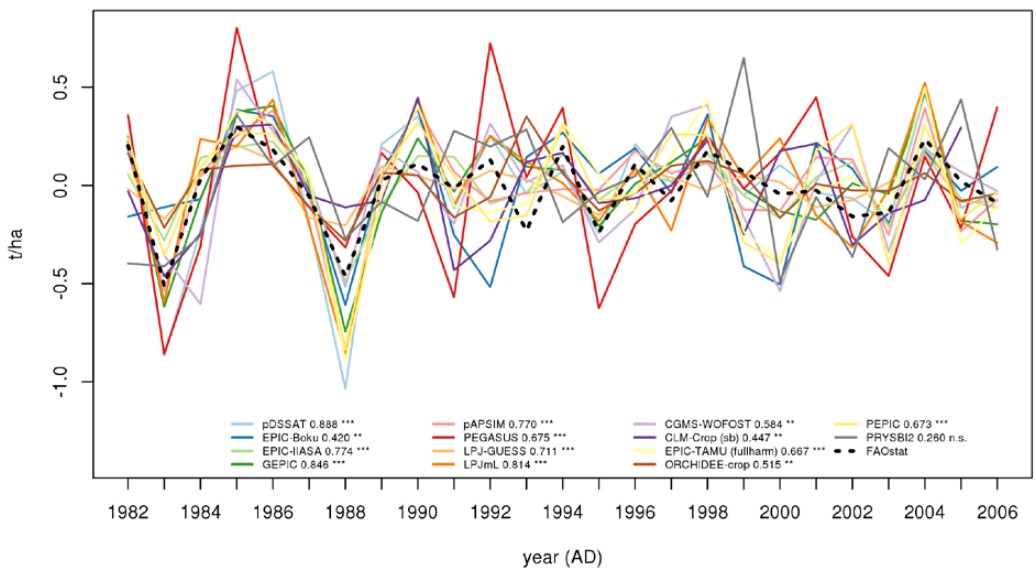


**Figure 1: Time series of GGCMI simulations (solid colored lines) and FAOstat reference data (dashed line) for maize after de-trending. Numbers in the legend next to model names indicate the Pearson correlation coefficient, asterisks indicate the p-values (*** for p<0.001, ** for p <0.05, * for p < 0.1, n.s. for not significant). This figure displays the 'default' setting, except for EPIC-TAMU, which only supplied the fullharm setting simulations (see Table 2). The (sb) flag indicates that the time series had been shifted backwards by a year to achieve a better match.**






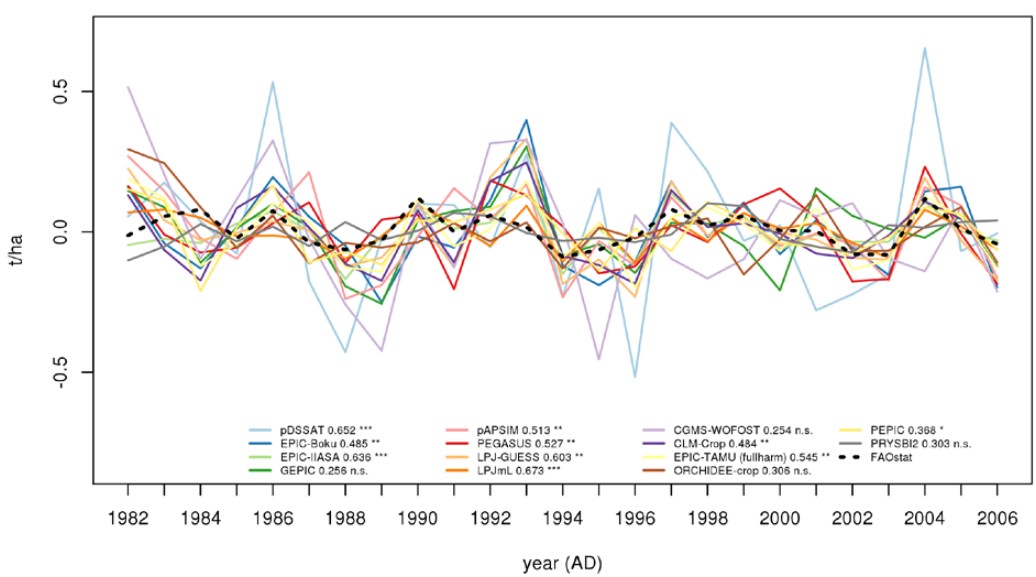


**Figure 2: As figure 1 but for wheat.**



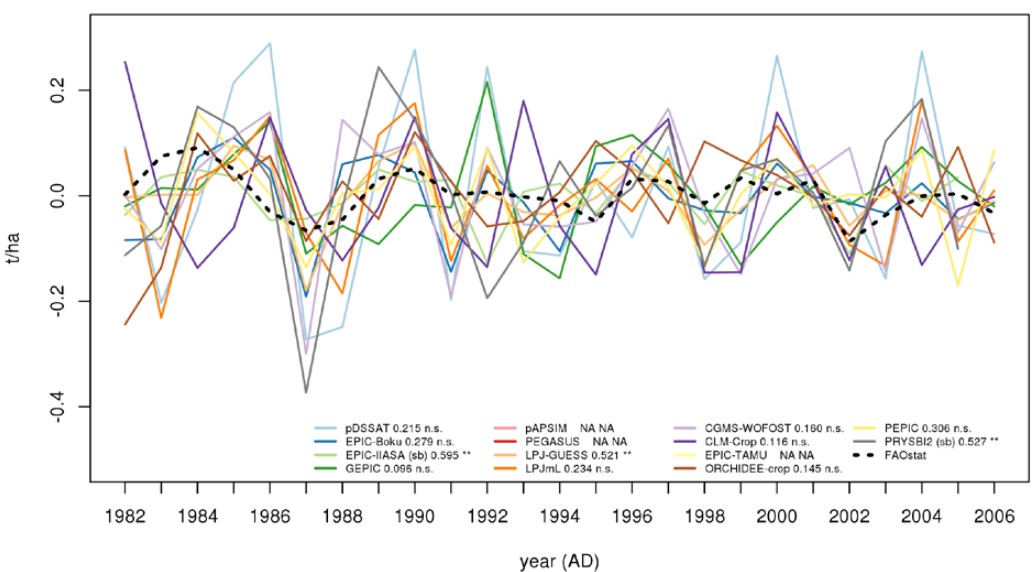


**Figure 3: As figure 1 but for rice. EPIC-TAMU, PEGASUS and pAPSIM did not supply data for rice.**





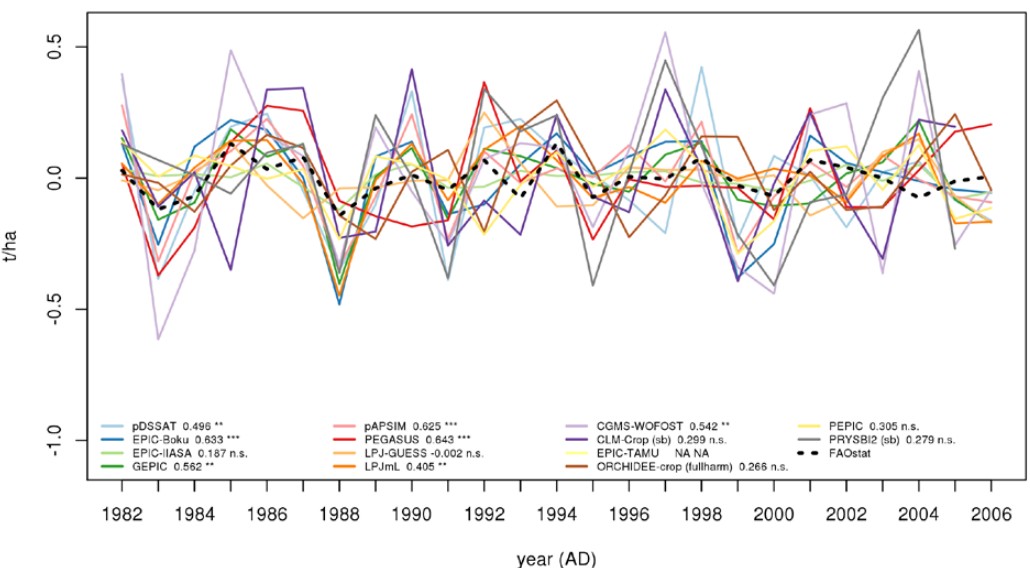


**Figure 4: As figure 1 but for soybean. EPIC-TAMU did not supply data for soybean.**





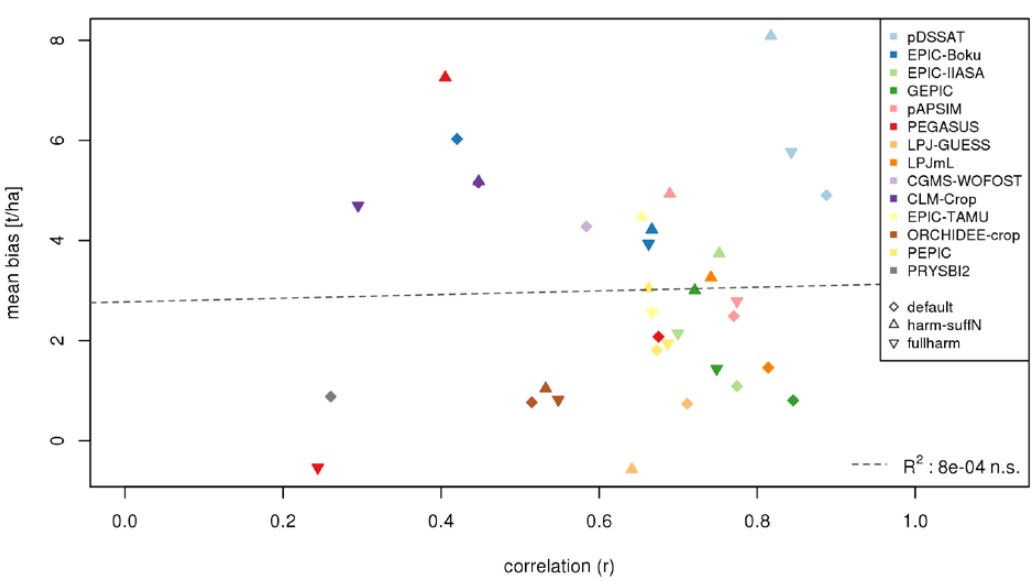


**Figure 5: relationship of global mean bias and time series correlation for maize across all GGCMs (colors) and harmonization settings (symbols). Dashed line indicates a linear fit, whose explanation power ($R^2$) is given in the right hand corner. Significance levels are as in figure 1.**






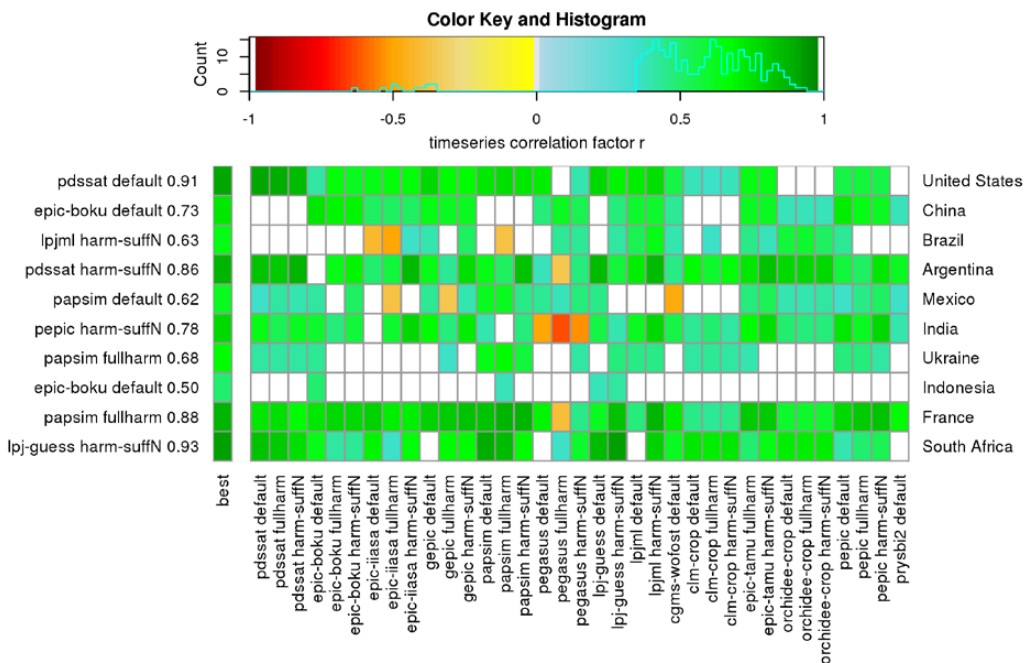


**Figure 6: time series correlation coefficients for the top 10 maize producer countries. Rows display the individual countries ordered by production; left-hand labels describe the best performing GGCM for that country and the correlation coefficients. White boxes indicate that correlations are not statistically significant. Each column displays individual GGCM x harmonization combinations, omitting all for which data is not available. The leftmost column displays the best correlation coefficient for each country (row), corresponding to the row labels on the left. Color legend key on top includes a histogram (cyan line) that shows the distribution of correlation coefficients across the ensemble and the top-10 producer countries, excluding the "best" column.**






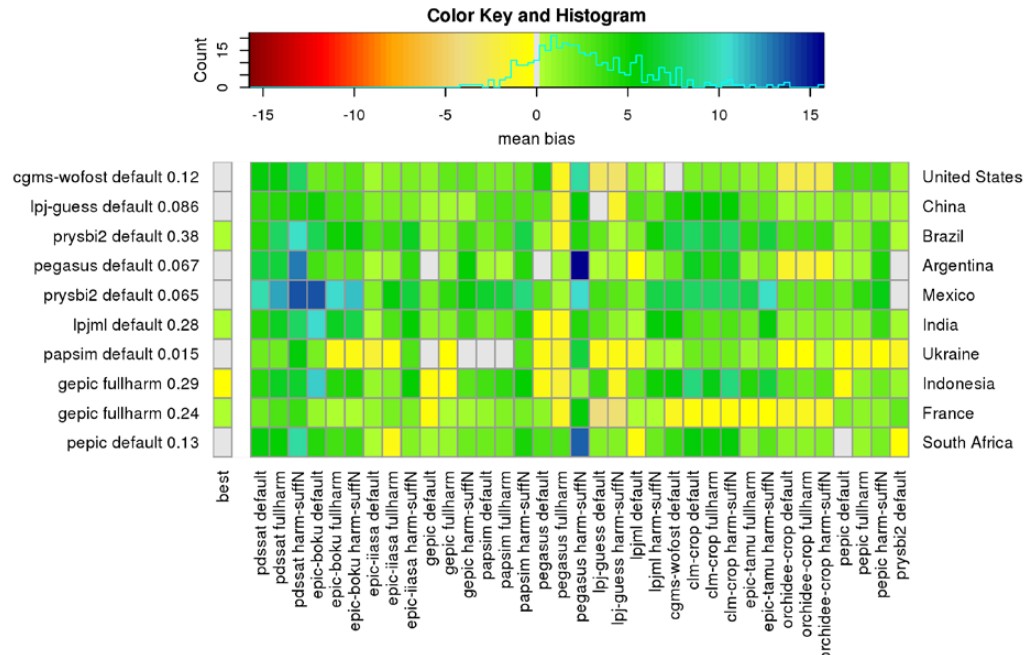


**Figure 7: As figure 6, but for mean bias (t/ha) of simulated yields for the top 10 producer countries for maize.**





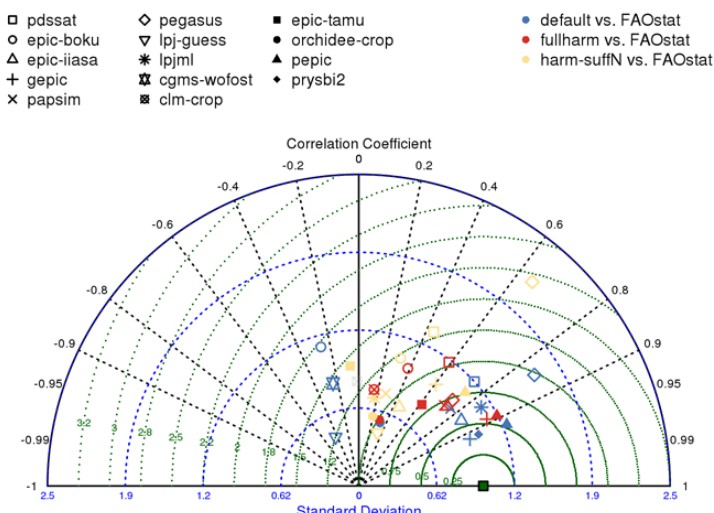


**Figure 8: Taylor diagram of maize yield simulations aggregated to national level against FAO statistics data after removing trends but preserving national mean yields. A perfect match with FAO statistics data would be at the dark green box on the x-axis, having a normalized standard deviation of 1 (distance to origin, blue contour lines) and a correlation of 1 (angle) as well as a centered RMSD of zero (green contour lines). Symbols represent the different GGCMs, colors indicate the harmonization setting. Non-significant correlations are shaded in lighter hues. Individual countries are weighted by their maize production according to FAOstat data (2014).**

939





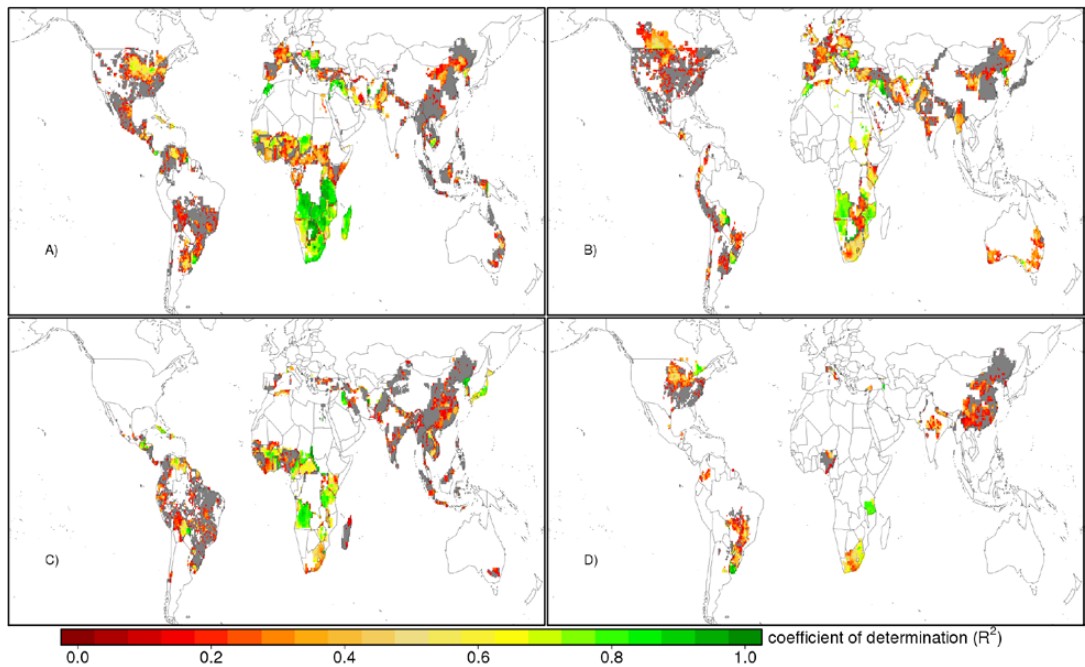

940

**Figure 9: Analysis of time series correlation between the two gridded yield reference data sets after removing trends via a moving average (see methods). Grey areas depict areas where there is no statistically significant correlation between the two data sets (p>0.1), white areas have no yield data for that crop in at least one of the two data sets. Panel A) shows coefficients of determination ($R^2$) for maize, B) for wheat, C) for rice, D) for soybean.**

945



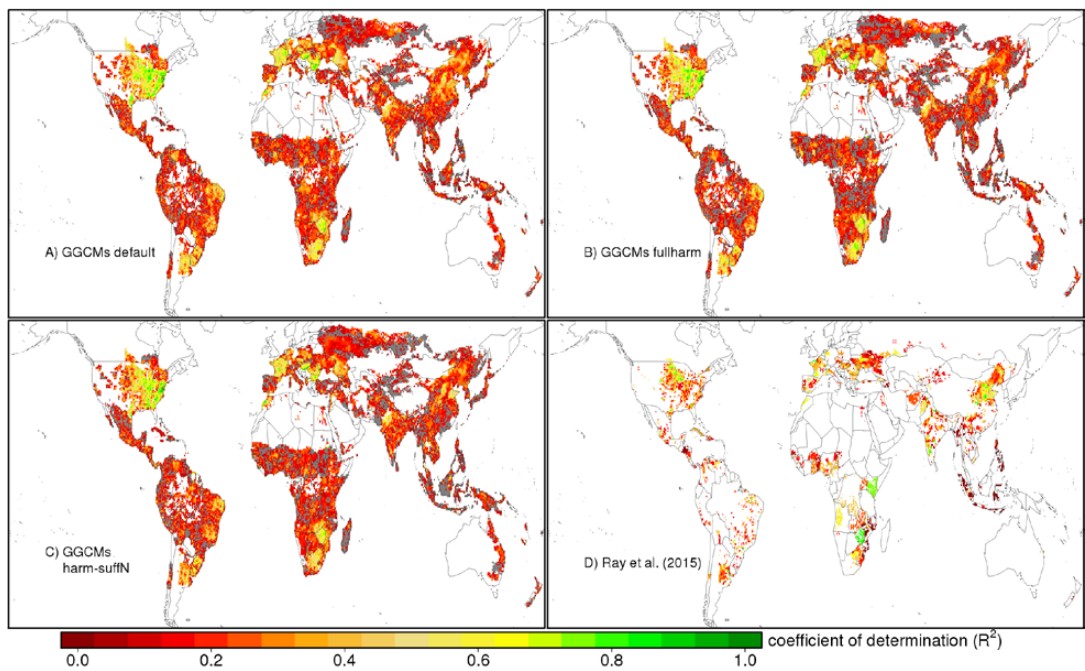

946

**Figure 10: Analysis of time series correlation between the GGCM ensemble simulations for maize (selecting best correlation across the GGCMs per grid cell) and the Ray2012 reference data set after removing trends via a moving average (see methods). Grey areas depict areas where none of the GGCMs finds a statistically significant correlation; white areas have no yield data for that crop in Ray2012 data sets. Panel A) shows coefficients of determination ($R^2$) for the *default* setting, B) for the *fullharm* setting, C) for the *harm-suffN* setting, and D) shows the original coefficients of determination as reported by Ray et al. (2015) for an ensemble of 27 regression models.**

953





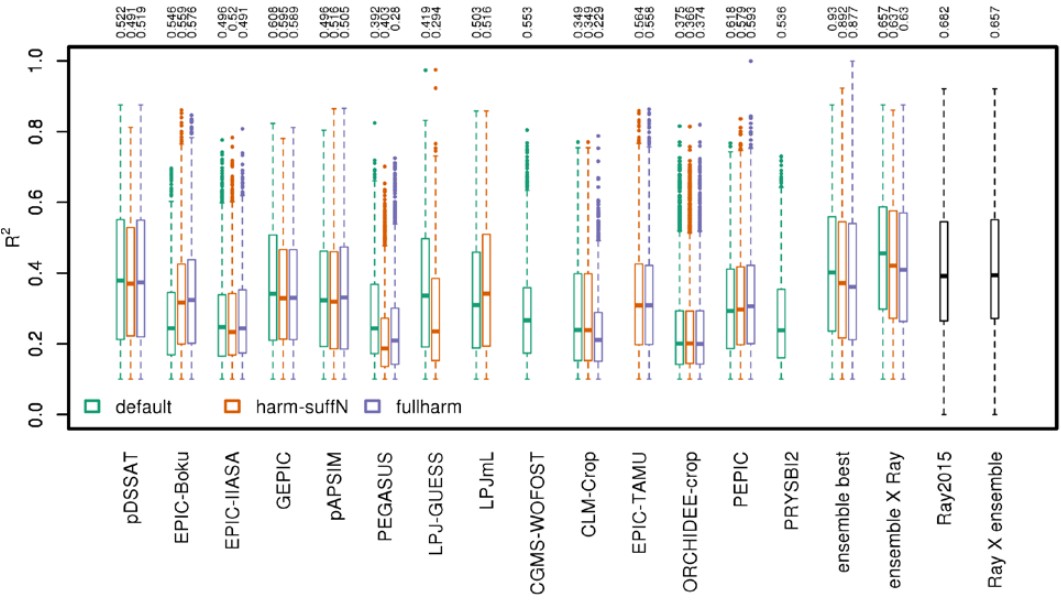

954

**Figure 11: Boxplot of R$^2$ distribution for each GGCM-harmonization setting for maize. Boxes span the interquartile range (25-75 percentiles); whiskers expand to the most remote value within 1.5 times the interquartile range. Values outside this range are considered outliers and are depicted as dots. The "ensemble best" shows the GGCMI skill-based (correlation coefficient) ensemble, "ensemble X Ray" is the same but only for those pixels where , and both are not independent from FAO national data also report significant correlations, "Ray2015" is the distribution of values as published by Ray et al. (2015), "Ray X ensemble" is as Ray2015 but only for the area where also the GGCMI ensemble reports significant correlation coefficients. The distribution is weighted by production, following the Ray2012 data set. Numbers at the top describe the fraction of the total harvested area for which significant correlations could be found, which ranges between 93% (ensemble best, *default*), 62% to 23% for the individual GGCMs and 68% for Ray et al. (2015).**

964