# Peer review of "Global Gridded Crop Model evaluation: benchmarking, skills, deficiencies and implications"

_Geoscientific Model Development, 2016_

## Referee Comment (RC1) · Anonymous Referee #1 · 19 Oct 2016

This is a well written paper. It proposes a framework to assess the performance of global gridded crop model. The framework will be a valuable asset for the research community. I think this paper has been submitted in a rush and I have some moderate concerns. 1) The authors claim that they will provide an online tool. It is a great idea but I wonder why not bring the evaluation system online before submitting the paper. 2) The paper cites some papers in preparation or under review which make it hard to refer to these papers. 3) There are too many figures and tables (with 45 figures in the supplemental file). And there are over 10 lines in some figures (Figure 1-4) that make the figures very busy. It is better to extract the key information and limit the number of figures if possible.

[Figure]

Specific comments

Line 169: What interpolation methods were used to disaggregate the daily data to sub-daily?

Line 177: The resolution of supplied input and harmonization data is 0.5 degree. The spatial scale of CLM-Crop, EPIC-IIASA and PRYSBT2 are 1 degree, 5 second and 1.125 degree. What is the method used to re-grid those data to 0.5 degree?

Line 180: "soy". However, in other place, the word is "soybean".

Line 215: delete the colon

---

## Referee Comment (RC2) · Anonymous Referee #2 · 21 Oct 2016

The development and evaluation of the global gridded crop models is a critical step in being able to provide an evaluation of the potential impacts of climate change on future global production. the authors have done a good job in explaining the process and the shortcomings in different models and approaches. This effort will set the stage for the next generation of improvements in crop models at all scales.

---

## Author Comment (AC1) · 14 Dec 2016

We thank the reviewers for their feedback. Our responses are inserted below , following their original comments.

Reviewer 1: This is a well written paper. It proposes a framework to assess the performance of global gridded crop model. The framework will be a valuable asset for the research community. I think this paper has been submitted in a rush and I have some moderate concerns.

Response: We thank the reviewer for the positive evaluation. We are sorry that we made the impression of having submitted the paper in a rush, which clearly was not

the case. Please find responses to your individual points below.

1) The authors claim that they will provide an online tool. It is a great idea but I wonder why not bring the evaluation system online before submitting the paper.

Response: The online evaluation tool is not the objective of the paper but an additional service to the modeling community. With the final publication of the paper, we'll make the online tool publicly available so that we can refer to this paper on the webpage. However, we now have included the URL of the tool (https://mygeohub.org/tools/ggcmevaluation), where access is currently restricted to the developers.

2) The paper cites some papers in preparation or under review which make it hard to refer to these papers.

Response: We assumed that these papers would have progressed sufficiently during the time our manuscript was under review. We will remove the references to Ruane et al. in prep. (which still is in prep) and update the references to Folberth et al. in prep. and to Prowollik et al. under review.

3) There are too many figures and tables (with 45 figures in the supplemental file). And there are over 10 lines in some figures (Figure 1-4) that make the figures very busy. It is better to extract the key information and limit the number of figures if possible.

Response: We agree that there are many figures and also a lot of information in the paper. This is why we have moved the majority of these into the supplement. The aim is to have sufficient information in the main document to convey the main message and to supply additional information for specific interests in the supplement. We cover the evaluation of 14 GGCMs for up to 4 crops each and establish a benchmark set for further model evaluation and future improvements with comparisons to reference data at three different aggregation levels. Therefore, also the extent of the study is very broad. We understand that it is the idea of GMD to supply all the space that is needed

to describe model evaluation in sufficient detail and don't feel that the content of our study is not concise enough. Also, to allow for individual model evaluation, we think that it is essential to show all individual models in one figure (as e.g. in figures 1-4), even though these are then busy.

Specific comments Line 169: What interpolation methods were used to disaggregate the daily data to sub-daily?

Response: ORCHIDEE-crop used an internal weather generator for the interpolation to sub-daily values, whereas CLM-crop created a 6-hourly weather input data set based on AgMERRA and the 6-hourly CRU NCEP data (Wei et al., 2014). This will now be explained in more detail in the supplement.

Line 177: The resolution of supplied input and harmonization data is 0.5 degree. The spatial scale of CLM-Crop, EPIC-IIASA and PRYSBT2 are 1 degree, 5 second and 1.125 degree. What is the method used to re-grid those data to 0.5 degree?

Response: CLM-crop used the model-internal re-gridding routine as described in the CLM 4.5 Technical Note (Oleson et al., 2013), PRYSBI2 simply averaged over all 0.5 grid cells within the 1.125 degree cells and EPIC-BOKU (not listed as 5 arc minute resolution in table S2, will be corrected) and EPIC-IIASA used the same climate and management input for all 5 arc minute cells (up to 36) within one single 0.5 degree grid cell. Thanks for pointing out that this is not described in sufficient detail and we will supply this information in the supplement and in Table S2.

Line 180: "soy". However, in other place, the word is "soybean".

Response: Will be changed to "soybean"

Line 215: delete the colon

Response: Will do.

Reviewer 2: The development and evaluation of the global gridded crop models is a

critical step in being able to provide an evaluation of the potential impacts of climate change on future global production. the authors have done a good job in explaining the process and the shortcomings in different models and approaches. This effort will set the stage for the next generation of improvements in crop models at all scales.

Response: Thank you.

References

Oleson, K. W., Lawrence, D. M., Bonan, G. B., Drewniak, B., Huang, M., Koven, C. D., Levis, S., Li, F., Riley, W. J., Subin, Z. M., Swenson, S. C., Thornton, P. E., Bozbiyik, A., Fisher, R., Heald, C. L., Kluzek, E., Lamarque, J.-F., Lawrence, P. J., Leung, L. R., Lipscomb, W., Muszala, S., Ricciuto, D. M., Sacks, W., Sun, Y., Tang, J., and Yang, Z.-L.: Technical Description of version 4.5 of the Community Land Model (CLM), NCAR Earth System Laboratory Climate and Global Dynamics Division, Boulder, CO, USANCAR/TN-503+STR, 2013.

Wei, Y., Liu, S., Huntzinger, D. N., Michalak, A. M., Viovy, N., Post, W. M., Schwalm, C. R., Schaefer, K., Jacobson, A. R., Lu, C., Tian, H., Ricciuto, D. M., Cook, R. B., Mao, J., and Shi, X.: The North American Carbon Program Multi-scale Synthesis and Terrestrial Model Intercomparison Project – Part 2: Environmental driver data, Geosci. Model Dev., 7, 2875-2893, 2014.
* * *

---

## Author Response (AR1)

**Response to reviewer comments (as also published in the online discussion)**

**Reviewer 1:**

This is a well written paper. It proposes a framework to assess the performance of global gridded crop model. The framework will be a valuable asset for the research community. I think this paper has been submitted in a rush and I have some moderate concerns.

We thank the reviewer for the positive evaluation. We are sorry that we made the impression of having submitted the paper in a rush, which clearly was not the case. Please find responses to your individual points below.

1) The authors claim that they will provide an online tool. It is a great idea but I wonder why not bring the evaluation system online before submitting the paper.

The online evaluation tool is not the objective of the paper but an additional service to the modeling community. With the final publication of the paper, we'll make the online tool publicly available so that we can refer to this paper on the webpage. However, we now have included the URL of the tool (https://mygeohub.org/tools/ggcmevaluation), where access is currently restricted to the developers.

2) The paper cites some papers in preparation or under review which make it hard to refer to these papers.

We assumed that these papers would have progressed sufficiently during the time our manuscript was under review. We will remove the references to Ruane et al. in prep. (which still is in prep) and update the references to Folberth et al. in prep. and to Prowollik et al. under review.

3) There are too many figures and tables (with 45 figures in the supplemental file). And there are over 10 lines in some figures (Figure 1-4) that make the figures very busy. It is better to extract the key information and limit the number of figures if possible.

We agree that there are many figures and also a lot of information in the paper. This is why we have moved the majority of these into the supplement. The aim is to have sufficient information in the main document to convey the main message and to supply additional information for specific interests in the supplement. We cover the evaluation of 14 GGCMs for up to 4 crops each and establish a benchmark set for further model evaluation and future improvements with comparisons to reference data at three different aggregation levels. Therefore, also the extent of the study is very broad. We understand that it is the idea of GMD to supply all the space that is needed to describe model evaluation in sufficient detail and don't feel that the content of our study is not concise enough. Also, to allow for individual model evaluation, we think that it is essential to show all individual models in one figure (as e.g. in figures 1-4), even though these are then busy.

Specific comments

Line 169: What interpolation methods were used to disaggregate the daily data to sub-daily?

ORCHIDEE-crop used an internal weather generator for the interpolation to sub-daily values, whereas CLM-crop created a 6-hourly weather input data set based on AgMERRA and the 6-hourly CRU NCEP data (Wei et al., 2014). This will now be explained in more detail in the supplement.

Line 177: The resolution of supplied input and harmonization data is 0.5 degree. The spatial scale of CLM-Crop, EPIC-IIASA and PRYSBT2 are 1 degree, 5 second and 1.125 degree. What is the method used to re-grid those data to 0.5 degree?

CLM-crop used the model-internal re-gridding routine as described in the CLM 4.5 Technical Note (Oleson et al., 2013), PRYSBI2 simply averaged over all 0.5 grid cells within the 1.125 degree cells and EPIC-BOKU (not listed as 5 arc minute resolution in table S2, will be corrected) and EPIC-IIASA used the same climate and management input for all 5 arc minute cells (up to 36) within one single 0.5 degree grid cell. Thanks for pointing out that this is not described in sufficient detail and we will supply this information in the supplement and in Table S2.

Line 180: "soy". However, in other place, the word is "soybean".

Changed to "soybean"

Line 215: delete the colon

Done.

**Reviewer 2:**

The development and evaluation of the global gridded crop models is a critical step in being able to provide an evaluation of the potential impacts of climate change on future global production. the authors have done a good job in explaining the process and the shortcomings in different models and approaches. This effort will set the stage for the next generation of improvements in crop models at all scales.

Thank you.

**List of all relevant changes to the manuscript:**

- Page 1: Removed NASA from affiliation 3, which is listed separately as affiliation 19
- Page 7: Inserted additional reference to supplementary where non-standard spatial and temporal resolutions are now described as requested by reviewer 1.
- Page 7: changed "soy" to "soybean" as requested by reviewer 1.
- Page 8: removed colon as requested by reviewer 1.
- Page 21: added the University of Chicago Research Computing Center to the acknowledgements.
- References:
    - Updated Folberth et al. in prep. to Folberth et al. 2016a (pages 5, 19, 23) and Folberth et al. 2016 to Folberth et al. 2016b accordingly (pages 4, 18, 23)
    - Updated Prowollik et al. under review to Porwollik et al. in press (pages 5, 10, 18, 25)
    - Updated URL to online tool, which will be released upon publication of this paper (pages 6, 21)
    - Removed reference to Ruane et al. in prep. which is still not available yet (pages 7, 17)
- We found a small bug in the data processing and updated figures 6, 8 and 11, as well as the corresponding figures in the supplement. None of these changes matters qualitatively, but only has small quantitative implications. We thus also updated the reported max correlation coefficient on page 13 from 0.45 to 0.42.

The marked-up manuscript version is supplied in the following (all changes in red).

[revised manuscript text omitted]

---

## Author Response (AR2)

**Response to reviewer comments (as also published in the online discussion)**

**Topical Editor:**

On the basis of the reviews and my own evaluation, I am pleased to tell you that the manuscript can be accepted after a technical correction on the line 53: GGCMS should be GGCMs.

We thank the editor Prof. Min-Hui Lo for the positive evaluation. We have corrected GGCMS to GGCMs in line 53 as requested.